# Oak Acorn Poisoning in Cattle during Autumn 2022: A Case Series and Review of the Current Knowledge

**DOI:** 10.3390/ani13162678

**Published:** 2023-08-20

**Authors:** Justine Eppe, Calixte Bayrou, Hélène Casalta, Dominique Cassart, Linde Gille, Margot Stipulanti, Jérôme Versyp, Arnaud Sartelet

**Affiliations:** 1Fundamental and Applied Research for Animals and Health Research Unit (FARAH), Clinical Department of Production Animals, Faculty of Veterinary Medicine, University of Liège, Quartier Vallée 2, Avenue de Cureghem 7A-7D, 4000 Liège, Belgium; calixte.bayrou@uliege.be (C.B.); hcasalta@uliege.be (H.C.); linde.gille@uliege.be (L.G.); margot.stipulanti@uliege.be (M.S.); jerome.versyp@uliege.be (J.V.); asartelet@uliege.be (A.S.); 2Fundamental and Applied Research for Animals and Health Research Unit (FARAH), Department of Animal Pathology, Faculty of Veterinary Medicine, University of Liège, Quartier Vallée 2, Avenue de Cureghem 6, 4000 Liège, Belgium; dominique.cassart@uliege.be

**Keywords:** cow, intoxication, oak, case report, pasture, kidney tubular necrosis, *Quercus*, Toxicology

## Abstract

**Simple Summary:**

Oak is widespread in Europe and can cause poisoning in grazing animals. Cattle are particularly susceptible. Herein, we describe seven cattle from three different farms admitted to the clinic for ruminants of the University of Liège for suspected acorn poisoning during autumn 2022. The clinical signs were vague. Blood analysis indicated renal failure. Of the hospitalized animals, five out of the seven had to be euthanized due to relapse. Lesions observed at necropsy were mainly digestive erosions and ulcerations, oedema and renal hemorrhages. Histopathological examination revealed necrosis of the renal tubules. Acorn poisoning is a serious disease with no specific antidote and no characteristic symptoms. Animals tend to be identified as sick late, when renal failure is already established. Farmers should be made aware of the prevention of this disease, especially in years when acorns are abundant. Furthermore, there is no antidote for this intoxication.

**Abstract:**

Oak poisoning is a known intoxication in grazing animals, but is slightly described in the literature. This case report describes 7 cattle from 3 different farms admitted to the clinic for ruminants of the University of Liège for suspected acorn poisoning in the autumn of 2022. The clinical signs were, anorexia, apathy with polyuria with low density. Further investigations led to the diagnosis of renal failure (blood urea 162 ± 88 mg/dL; blood creatinine 12 ± 4 mg/L). Supportive treatment, based on infusions (NaCl 0.9%) and electrolyte rebalancing, was administered and renal values were assessed every 24–48 h. Of these animals, 5/7 were euthanized. At necropsy, digestive erosions and ulcerations, oedema and renal hemorrhages, between the pyloric/caliceal cavity and the medulla were observed. Histopathological examination revealed necrosis of the renal tubules. The renal values of the two remaining animals were reduced, their general condition improved, and they were discharged. Acorn poisoning is a serious disease with no specific antidote or characteristic symptoms. Animals are identified as sick too late, when renal failure is already established. Farmers should be made more aware in order to prevent exposure, especially in years when acorns are abundant. Furthermore, there is no antidote for this intoxication.

## 1. Introduction

Oak (*Quercus* spp.) is the most common tree species in northern Europe [1,2]. It grows both in forests and on the edge of meadows. Many species of oaks in north America and Europe are toxic [3]. Its buds, twigs, leaves and acorns are toxic for grazing animals [2,4,5,6].

Acorn poisoning in ruminants is caused by a direct irritant action of tannic acids on the gastrointestinal mucosa (rumenitis, ulcers, hemorrhages), and indirectly via the degradation products of tannic acid through rumen digestion in gallic acid and pyrogallol [4,7,8]. Sudden death in cases of massive ingestion may occur at this stage [2,4,9]. The absorption of gallic acid and pyrogallol causes—through a strong inflammation stimulus on the glomeruli and the proximal segment—a multifocal necrosis of the renal tubules with the creation of protein cylinders [3,7]. The precise mechanism of tannin compounds is still poorly understood, but it is speculated that they are able to plug the loop of Henle by entering the cell and forming a precipitate by combining with the cell membrane or nutrients, or that the simple irritant effect of tannins causes lesions in the organs where they accumulate most [2]. This leads to chronic renal failure several weeks after ingestion, with all its consequences: ascites, hydrothorax, perirenal, retroperitoneal and subcutaneous edema [4,7,9]. The lesions observed on the kidneys allow acorn poisoning to be diagnosed [3,7].

Most of the available literature on acorn poisoning in cattle has been set for a long time [6,7,10]. The clinical signs are loss of appetite, constipation followed by diarrhea, colic or emaciation [2,7,9]. The examined cattle show apathy. The heart rate may be within normal limits, but both tachycardia and bradycardia have been reported [6,7]. The respiratory rate may also be increased [6,7]. In the digestive system, inappetence and increased thirst are described. The rumen shows reduced motility or atonic, and the digestive sounds are weak or inaudible on auscultation [2,7]. Another observation is the frequent emission of a large quantity of very clear urine, with low specific gravity, containing glucose and albumin. The urinary pH is slightly acidic (mean = 6.7), certainly compared to the usual pH of 7–8 [7,11].

Blood biochemistry revealed uremia, creatinemia, dehydration and hypocalcemia [2,7,9]. Treatment generally consists of intensive supportive therapy: intravenous and subcutaneous infusions, correction of electrolyte disorders, glucose supplementation and charcoal administration. Renal parameters are monitored for a possible reversal of the renal failure [2,7]. This reversal is associated with an improvement in the animal’s general condition, with a recovery of its appetite.

The prognosis of this intoxication is poor, with few (15–25%) affected animals recovering [7]. In view of the poor prognosis of this intoxication, the prevention and risk of acorns in a meadow must be emphasized and mitigated [7].

Although the oak is a tree commonly found in our grasslands, full case report descriptions, with supporting further laboratory analysis, are rare in the literature [4,9]. This case report provides a unique and contemporary description of seven cases of acorn poisoning in cattle received at the clinic for ruminants of the University of Liège (CRU Liège) in the autumn of 2022.

## 2. Cases Information

Between 13 October and 1 November 2022, the CRU Liège received seven young cattle, from three different farms, referred by the field veterinarians for suspected oak acorn poisoning. These animals were between 9 and 4 months old, all Belgian Blue Cattle breed (BBCB), with five females and two males. 

All the animals presented at the CRU Liège had in common that they were grazed 3 weeks to 1 month before the onset of the clinical signs in meadows bordered by oak trees. The animals all showed apathy, anorexia, ruminal stasis and absence of feces. Some animals (4/7) also had an episode of hyperthermia. Their condition had not improved despite the treatments administered by the veterinarians (broad spectrum antibiotics, non-steroidal anti-inflammatories, vitamins A, E, zinc, selenium and magnet). 

For two of the calves, blood analysis was performed by the referring veterinarian on a pooled blood sample, revealing renal insufficiency (urea = 341 mg/dL; normal range (No): 10–25 mg/dL; creatinine = 24.87 mg/dL; No: 0.4–1 mg/dL [12]), increased liver values (GLDH = 52.3 IU/L; No < 30 IU/L); Gamma GT (63 IU/L; No < 39 IU/L [12]) and hypocalcemia (67 mg/L; No = 83–104 mg/L [12]). 

## 3. Clinical Findings

At the clinical examination, one calf arrived in sternal recumbency, while the others were standing (Table 1). All were lethargic. Some of the animals were hypothermic with a rectal temperature below 38.5 °C and one was hyperthermic (39.7 °C). Some of the animals showed signs of dehydration (capillary refill time greater than 2 s, skin folds greater than 3 s and enophthalmia). None of the animals had adenomegaly. At the cardiorespiratory auscultation, a heart murmur (systolic; 4 to 5/6 gradation) was found on two out of the seven cattle. One of these two cattle also had an arrhythmia. Abdominal auscultation revealed a decrease in digestive sounds. During the examination, the calves frequently urinated very clear urine.

## 4. Diagnostic Assessment

For two of the seven animals (n°5,7), renal failure was already confirmed before their arrival. In view of the history and clinical examinations, acorn poisoning remained as the first diagnostic hypothesis. Other differentials may be suspected, such as other intoxications or poisonings (plants with soluble oxalates, ethylene glycol, etc.) or any other causes of renal impairment (pyelonephritis, *Leptospira* spp., etc.).

Due to financial constraints and the poor prognosis, not all complementary examinations were carried out on all of the animals (Table 2). All of the cattle had a blood test for both urea and creatinine (plasma, Catalyst One^®^, Idexx, Westbroock, ME, USA). Blood gaz analysis (Vetstat^®^, Idexx, Westbroock, ME, USA was conducted on some of the animals (4/7)). Furthermore, L-lactate was determined on three of the animals (n°1, 2, 3) (Accutrend^®^ Plus, Roche Diagnostics International AG, Rotkreuz, Switzerland). A complete hematology and biochemistry (biochemistry: Catalyst One^®;^ hematology: Procyte Dx^®^, Idexx, Westbroock, ME, USA) was performed on three of the seven animals. 

The animal (n°7) presenting with arrhythmia received additional blood gaz and ion analysis to evaluate its potassium level. The potassium level was within the normal range for this calf. 

The obtained results for the additional animals revealed hyperkalemia in one out of the four, hyponatremia and hypochloremia in all four and a reduced bicarbonate concentration and metabolic acidosis in three out of the four. 

In terms of haematology and biochemistry, all seven animals showed severe uremia and creatinemia, three-thirds had neutrophilia, basophilia, hyperphosphataemia and hypocalcaemia, two-thirds had leukocytosis and hyperlactatemia and one-third had monocytosis.

Urinary analysis using a dip stick (Combur 10 Test, Roche Diagnostics Gmbh, Deutschland) was performed on one of the animals (n°3), which revealed glycosuria, acidic pH at 6 and proteinuria. The urine density was 1010 g/mL.

The combination of the above-described analysis led to a diagnosis of severe renal failure, with more or less electrolytic repercussions. The most likely diagnosis was acorn poisoning. Unfortunately, there is no specific diagnostic test for this disease. In view of the very severe kidney damage and the suspected diagnosis, the prognosis for the animals was poor.

## 5. Therapeutic Intervention

The basis of the treatment administered was a 0.9% NaCl intravenous therapy at a rate of 10 mL/kg/h. If oedema developed, the rate of infusion was reduced to 5 mL/kg/h and intravenous diuretics (Furosemide, 1 mg/kg, Dimazon^®^, Intervet International, MSD Animal, Bruxelles, Belgium) were added to the treatment. Depending on the treatment previously administered by the referring veterinarians and the analysis carried out on the animals, some animals received additional oral rehydration (DrenchDig^®^, Savetis, Quévert, France), rumen digestion stimulants (Rumiphyt^®^, Savetis, Quévert, France) calcium (Parpumag 30%^®^, Dechra, Northwick, United Kingdoms) and/or potassium chloride. Calcium and potassium chloride were distributed based on the ionic losses in the blood analysis results.

## 6. Follow Up 

The urea and creatinine were measured every 24 to 48 h depending on the case. If the urea and creatinine levels did not reduce to the measurable range, which is associated with a non-improvement of the general condition (anorexia, persistent depression despite treatment) or even a worsening via the appearance of subcutaneous oedema, euthanasia of the animals was recommended. 

During hospitalization, five of the seven cattle rapidly developed sub-ventral and pulmonary oedema (12 h after infusion) (Figure 1). These animals showed tachycardia, tachypnea and crackles on pulmonary auscultation. The infusion was decreased to 5 mL/kg/h and intravenous diuretics were added to the treatment of these animals.

For four of the seven cattle, no improvement, deterioration and/or unchanged renal parameters led to the decision to euthanize them 48 h after arrival.

Appetite returned in three of the seven cattle (n° 3, 4, 7) after 12 h of infusion, and the kidney values decreased after 24–48 h in two of these animals (cattle n° 3, 7). Within these three animals (cattle n° 3, 4, 7), two did not show sub-ventral edema. The treatments were continued for these animals while waiting for the renal parameters to return to a normal value (follow-up in Table 3). For cattle n°4, the renal values never decreased, and the sub-ventral oedemas persisted. He became anorexic again on the 5th day of hospitalization and was euthanized.

## 7. Necropsy Findings

The five euthanized calves were necropsied. They were all in good shape. All had more or less severe generalized oedemas: subcutaneous cavitary oedema, hydrothorax, hydroperitoneum, hydropericardium. Some cavitary oedemas contained ten liters of fluid. 

Lesions of the digestive tract were visible in all of the animals with congestion of the oesophagus, ulcerative abomasitis or small haemorrhagic foci (1–2 mm diameter) in the small intestine wall. The ulcers observed on the abomasal mucosa measured 6–7 mm by 3–4 mm and were located on top of the abomasal folds. The intestinal contents were liquid.

The other lesion systematically observed was a haemorrhagic zone at the junction between the medulla and the calyces/pyelic cavities in the renal interstitial tissue (Figure 2). These lesions were severe and involved both kidneys.

Microscopic examination of the kidneys revealed multifocal tubular necrosis with granular cylinders and foci of subacute interstitial nephritis (Figure 3). Microscopic lesions were concentrated in the renal tubules, with the glomeruli appearing intact. A different level of damage to the renal tubules was observed on the different histological slides performed on the animals, but this lesion was found in all of them. The tubules were dilated, their epithelium was absent and they were filled with necrotic debris.

## 8. Outcome

Two cattle (n° 3 and 7) were discharged after several days of IV fluidotherapy (15 and 7 days, respectively). Both had their urea returned within the normal range, but their creatinine levels remained above the reference value (creatinine 5.2 and 2.8 mg/dL, respectively).

A follow-up 6 months later showed that the young cattle grew normally and were in good general health. For cattle n° 7, its fattening condition allowed it to be sold and sent to the slaughterhouse. There is no information on the outcome, nor its carcass weight. For cattle n° 3, a blood test was carried out. It revealed a normal uremia (140 mg/dlL), but the creatinine remains (2.2 mg/dL) slightly above the reference value. The heifer was stunted, with a chest circumference of 166 cm, the estimated weight being 370 kg (according to the formula [15]. Her clinical examination was within the normal range.

## 9. Discussion

The prevalence of exposure to acorns and subsequent poisoning depends on the year, the season, the age of the oak, the species, the maturity of the acorns and the weather conditions [3,6,7]. Summer drought, winds or windstorms are climatic risk factors for finding a lot of acorns on the meadow floor [3,8,16,17]. Further, it is reported that in some years, more acorns are produced by oaks [7]. Younger oaks also contain more gallic and tannic acids, with this concentration decreasing with the age of the oak [3]. Finally, some oak species are naturally more toxic, such as the English oak (*Quercus robur*) [3]. All of these considerations may explain why there is not the same frequency of acorn poisonings every year, although cases are reported every year at the Laboratory of Toxicology of the Faculty of Veterinary Medicine at Ghent University (LTGU) [17]. 

The toxic molecules contained in the leaves, acorns, blossoms and buds of the oak are gallic and tannic acids [3,4,8]. Oak leaves are most toxic in early spring, along with the buds and blossoms. The leaves tend to lose their toxicity with age, so that the brown leaves that fall in autumn are much less toxic. For acorns, the risk is greatest during the autumn period [6]. Acorn contains a high level of pyrogallic acids and are more toxic when immature and green. A mature acorn is considered to have little or no toxicity [3,7]. 

Acorn poisoning rarely affects grazing animals other than cattle, such as horses [3], goats, sheep [3] and camelids [18]. 

Not all cattle seem to be interested in acorns, and intoxication is often due to excessive curiosity on the part of young animals (0–3 years) [3,7,10], as was probably the case here, or the scarcity of grass in cases of drought [9,16]. Once the animals have tasted the acorns, buds or leaves, they may develop a craving for more [2]. Most of the affected animals were females (5/7). To the best of our knowledge, there is no information on the sex of animals poisoned by acorns. This sex ratio is probably due to the fact that males are principally fattened without access to the outside, whereas young females grow up in the meadows from April to November after one year of age. 

In this case report, the episode followed a dry summer. The strength of this case report lays in the large amount of recent information collected on acorn poisoning cases, as well as the follow-up of these hospitalizations and the necropsy findings as little information is available in the current literature. Unfortunately, there is no clear protocol for the management of these clinical cases due to economical restraints, in some cases combined with the poor literature available, which leads to heterogeneity in data collection. 

For chronic intoxication, clinical signs appear 1 to 3 weeks after grazing, which corresponds to what has previously been described [6,7]. Not very suggestive clinical signs, such as apathy, anorexia and an absence of feces, were reported. Loss of appetite is the first sign reported in the literature [6,7]. Constipation, a direct effect of tannic acid, followed by diarrhea with mucus and blood 1 to 8 days later, are also described [3,6]. The animals suffer kidney failure, which can be irreversible [19]. High water consumption and frequent emission of large amounts of urine have been reported [3,6]. In pregnant animals, abortion can also occur [2]. The animal stops gaining weight, stops growing, may be anemic and has declining subcutaneous oedema as a consequence of this renal failure [3,6]. In this case, the clinical signs related to renal failure were well observed, but not those related to the gastrointestinal tract, probably because the animals were already well advanced in the disease and anorexic. The hyperthermia that was also reported in four of the seven animals was not consistent with the symptomatology of acorn poisoning and was probably caused by another concomitant disorder.

The blood tests showed signs of renal failure: increased blood urea (7/7), creatinine (7/7), phosphate (3/3), potassium (1/4) and decreased blood chlorine (4/4) and calcium (3/3). Some of the animals were in metabolic acidosis (3/4). Dilution before the analysis was not systematically carried out when the values were above the measuring range in order to reduce the owner’s costs. The decrease in ions and bicarbonate observed in some of the animals could be related to anorexia. Finally, some of the animals showed signs of inflammation on the haematology (leukocytosis: 2/3; neutrophilia: 3/3; monocytosis: 1/3). These results are consistent with what is described in the literature [7,20]

Tannic acids precipitate proteins on the cell membranes in the digestive tract, and cause erosion and ulcerations, in turn altering the absorption (injured mucosa) [5,19]. It can also increase vascular permeability [2] and edema, ascites, pleural or pericardial effusions may be observed, some containing several tens of liters of fluid [6]. The liver can also be affected [3,19]. The most characteristic lesion of this intoxication is the hemorrhage observed between the medulla and the calyces/pyelic cavity of the kidney. Microscopic examination allows a definite diagnosis to be made when tubular necrosis is observed [5,6]. The toxic compound causing these renal tubular necroses is gallic acid [19]. The suspected pathogenesis is a combination of tannin components with cell membrane proteins or nutrients, causing the cell to become necrotic, with these protein compounds then accumulating in the renal tubules [2]. All of the cattle examined at necropsy had these lesions. The microscopic examination provided a definitive diagnosis of these animals, revealing tubular necrosis. For cattle n°3, there is still doubt as to whether he suffered from acorn poisoning, as none of the animals in his herd have been autopsied and diagnosed, and there is no in vivo test to confirm acorn poisoning.

The prognosis is poor, with a fatal outcome in 75–85% of cases, mainly due to the non-reversibility of renal failure [7]. The number of acorns and the severity of renal failure probably play a role in the prognosis of this disease. One source indicates that eating 1 kg of green acorns for 15 days would be sufficient to induce intoxication in cattle [21]. The prognosis is poor with a urea blood level above 100 mg/dL, and hopeless above 300 mg/dL [2,21]. Regarding the poor prognosis of this poisoning, for economic and welfare reasons, it would be better to euthanize the animals and not attempt medical treatment. In this case report, two of the seven animals were discharged. The decrease in the renal values was already noticeable 24 h after the start of fluid therapy. Despite a satisfactory recovery, the surviving animals still retained their stunted growth, although they no longer relapsed at the renal level. A realistic protocol for the management of acorn poisoning could be to administer intravenous fluid therapy to the animals for 24 h, check their urea and creatinine, and if these reduce, continue fluid therapy until normal renal values are restored. For those animals whose renal values do not change, their case should be considered hopeless and euthanasia is recommended in these chronic cases.

There is no specific antidote for the tannins in oak, but several studies have compared diets containing salts (calcium hydroxide, hydrated lime) and/or vegetable oil to prevent this poisoning [6,8,16]. Trials with rations containing these additives are promising for limiting acorn poisoning, but this means supplementing animals that are on grassland with a feed that not all of them are willing to eat [6]. Once clinical signs are present, it is advisable to carry out an intensive supportive treatment based on intravenous or subcutaneous infusions and the correction of electrolyte disorders (calcium, potassium) [7]. A clinical case on a zebu (*Bos taurus indicus*) describes an effective treatment of acute renal failure caused by this intoxication with hemodialysis [20]. However, hemodialysis is too expensive to be used to treat production animals.

## 10. Conclusions

Exposure to acorns can have severe repercussions, which can cause serious after-effects several weeks after ingestion. The absence of an effective treatment reinforces the need to insist on prevention. Suspected cases are observed every year. Once clinical signs appear, it is often too late to save the animal. First, digestive lesions are present, followed by renal failure a few weeks after, with frequently irreversible lesions. Symptomatic treatment is the only solution. Uremia and creatininemia monitoring seems to be the solution for decision-making. However, this intoxication must be put into perspective, because it depends on many parameters, such as the type of oak, the climatic conditions and the year. As with all poisonous plants, meadows should be examined before the animal grazing period and should be avoided at the end of the season on pastures with a lot of acorns on the ground. Future studies should focus on the development of a diagnostic test, given the non-specific clinical signs and consequences of this disease.

## Figures and Tables

**Figure 1 animals-13-02678-f001:**
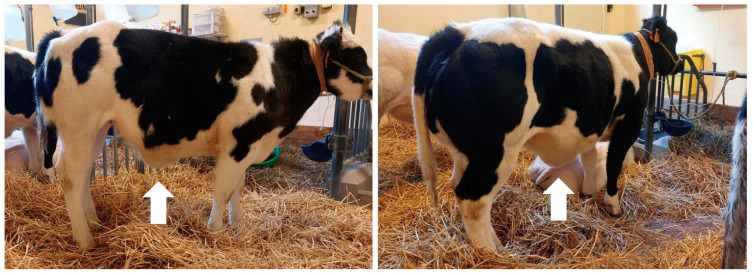
Sub-ventral oedema (arrow) of two cattle (n° 5 and 6).

**Figure 2 animals-13-02678-f002:**
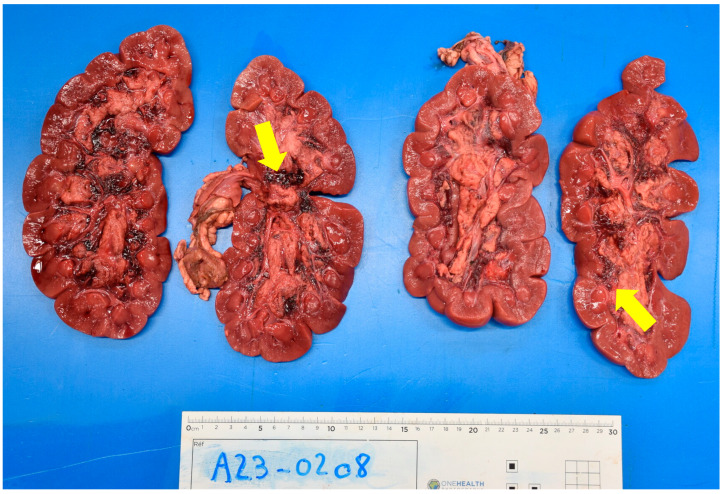
Kidneys of calf n° 6. Hemorrhagic areas can be seen at the junction between the medulla and the calyces/pyelic cavities (yellow arrows) as well as a pitted appearance of the cortex.

**Figure 3 animals-13-02678-f003:**
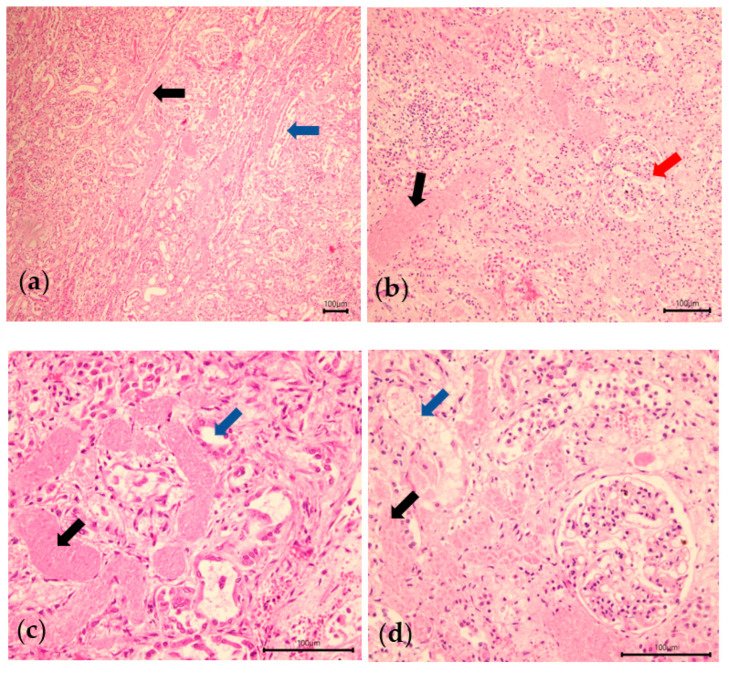
Histological images of kidney sections stained with hematoxylin-eosin at 100× (**a**), 200× (**b**), 400× (**c**,**d**) magnification. In (**a**), the extent of tubular necrosis and protein cylinders in the renal tubules can be seen. The black arrows show tubules containing hyaline necrosis, protein cylinders. The blue arrows show intact renal tubules. In (**b**) another image of hyaline tubular necrosis (black arrow) with a glomerulus (red arrow). In (**c**), section in necrotic tubules (black arrows), and in healthy tubules (blue arrows). In necrotic tubules, the tubular epithelium is no longer visible at all, and the lumen of the tubule is completely obstructed by necrotic material. In (**d**), another comparison between a healthy tubule (blue arrow) and a necrotic tubule (black arrow).

**Table 1 animals-13-02678-t001:** Clinical examination parameters observed on the 7 animals, compared with the normal range [12]. Values outside the norm are shown in bold.

Cattle	1	2	3	4	5	6	7	Reference Value
Sex	male	female	female	female	female	female	male	-
Weight (kg)	200	202	335	275	210	290	127	-
Age (month)	10	9	17	8	7	8	5	-
Respiratory rate (respiration/min)	**12**	24	**54**	**112**	40	32	24	20–40
Rectal temperature (°C)	**36.7**	**38.1**	**38**	**38.2**	**38.1**	38.9	**39.7**	38.5–39.5
Heart rate (Beats/min)	76	**52**	72	**108**	**108**	**110**	72	60–84
Capillary refill time (s)	**3–4**	<2	<2	<2	<2	<2	**3**	<2
Skin folds (s)	**5**	**3**	**3**	**3**	<2	<2	<2	<2
Enophtalmia (mm)	**3**	**2**	**2**	0	0	0	0	0
Mucous membranes	**Pale**	Pink	Pink	Pink	Pink	Pink	**Congestive**	Pink
Lymph nodes	Normal	Normal	Normal	Normal	Normal	Normal	Normal	Normal

**Table 2 animals-13-02678-t002:** Results of blood tests performed on cattle, compared to reference standards [12,13,14]. Values outside the norm are shown in bold. (-) indicates that the test has not been carried out for this animal.

Cattle	1	2	3	4	5	6	7	Reference Value
Urea (mg/dL)	**>130**	**>130**	**321**	**>130**	**>130**	**>130**	**124**	10–20
Creatinine (mg/dL)	**>13.6**	**>13.6**	**3.63**	**>13.6**	**>13.6**	**>13.6**	**>13.6**	0.4–1
Lactate (mmol/L)	**2.6**	1.3	**4**	-	-	-	-	<2
Blood pH	**6.96**	**7.24**	**7.22**	-	-	-	7.35	7.35–7.50
HCO3− (mmol/L)	**10.6**	**17.7**	**13.6**	-	-	-	25.3	24–34
Na (mmol/L)	**<100**	**122**	**129**	-	-	-	**124**	134–145
K+ (mmol/L)	**8.4**	1.9	2.3	-	-	-	1.4	3.9–5.3
Cl− (mmol/L)	**73**	**89**	**94**	-	-	-	**85**	94–105
Hematocrit (%)	29.5	27.1	23.6	-	-	-	-	22–33%
Hemoglobin (g/dL)	10.3	9.8	9	-	-	-	-	8–15
Leucocytes (×10^9^/L)	**15.82**	**15.19**	10.05	-	-	-	-	4–12
Neutrophiles (×10^9^/L)	**8.54**	**11.49**	**6.4**	-	-	-	-	0.6–4
Lymphocytes (×10^9^/L)	6.54	2.85	2.64	-	-	-	-	2.5–7.5
Monocytes (×10^9^/L)	0.63	0.75	0.94	-	-	-	-	0.025–0.84
Eosinophiles (×10^9^/L)	0.09	0.04	0.03	-	-	-	-	0.0–0.24
Basophiles (×10^9^/L)	**0.02**	**0.04**	**0.03**	-	-	-	-	0.00–0.02
Platelets (K/µL)	508	579	426	-	-	-	-	100–800
P (mg/L)	**>161**	**157**	**154**	-	-	-	-	42–77
Ca (mg/L)	**48**	**49**	**48**	-	-	-	-	83–104
Total proteins (g/L)	68	77	**91**	-	-	-	-	70–85
Albumin (g/L)	35	40	42	-	-	-	-	32–42
Gamma-glutamyltransferase (IU/L)	28	26	38	-	-	-	-	11–39
Cholesterol (g/L)	0.23	0.81	0.98	-	-	-	-	0.73–2.8

**Table 3 animals-13-02678-t003:** Follow up renal values, urea (U) and creatinine (C) in mg/dL of cattle.

n°	DAY 1		DAY2		DAY3		DAY5		DAY10		DAY16	
	U	C	U	C	U	C	U	C	U	C	U	C
1	>130	>13.6										
2	>130	>13.6										
3	321	3.63	18	1.7					29	1	10	5.2
4	>130	>13.6	>130	>13.6	>130	>13.6	>130	>13.6				
5	>130	>13.6	>130	>13.6								
6	>130	>13.6	>130	>13.6								
7	124	>13.6	80	11.1	59	7	22	3.9	6	2.8		

## Data Availability

All data are available in the manuscript.

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
