# Peer review of "Oak Acorn Poisoning in Cattle during Autumn 2022: A Case Series and Review of the Current Knowledge"

_animals, 2023, doi:10.3390/ani13162678_

Round 1

Reviewer 1 Report

my comments are :

My comments are that the authors demonstrate a higher incidence of intoxication in females compared to males. The authors should better specify this aspect in the discussion.

The increase in neutrophil leukocytes could be associated with an increase in a pro-inflammatory state ?

Author Response

REVIEWER 1

Dear Reviewer 1,

Thank you very much for your review.

My comments are that the authors demonstrate a higher incidence of intoxication in females compared to males. The authors should better specify this aspect in the discussion.

We have added a few sentences on the subject in lines 349-353.

The increase in neutrophil leukocytes could be associated with an increase in a pro-inflammatory state ?

Yes, we talked about this on lines 379-381.

Best regards,

Reviewer 2 Report

The article by  Justine Eppe, Calixte Bayrou, Hélène Casalta, Dominique Cassart, Linde Gille, Margot Stipu lanti, Jérôme Versypand Arnaud Sartelet entitled “Oak Acorn Poisoning In Cattle During Fall 2022: A Case Series and Review of the 2 Current Knowledge”, propose to investigate the Oak Acorn poisoning in cattle.

 I recommend that the paper be accepted with minor revision:

a)     The authors should check for typographical and grammar error the entire manuscript (space, page lines etc…)

b)    The references included in the manuscript, with the exception of a few, are older than 5 years. Please enter the most recent references. 

c)     Authors, please add more references to the toxic effects of Oak Acorn; many symptoms have been listed, but all with a single reference. 

d)    The authors are requested to expand the introduction. 

Author Response

The article by  Justine Eppe, Calixte Bayrou, Hélène Casalta, Dominique Cassart, Linde Gille, Margot Stipu lanti, Jérôme Versypand Arnaud Sartelet entitled “Oak Acorn Poisoning In Cattle During Fall 2022: A Case Series and Review of the 2 Current Knowledge”, propose to investigate the Oak Acorn poisoning in cattle.  I recommend that the paper be accepted with minor revision:

Dear Reviewer 2,

Thank you very much for your reviewing.  You will find all the changes in yellow in the text.

  1. a) The authors should check for typographical and grammar error the entire manuscript (space, page lines etc…)

We went through the entire document to find these typographical errors, which should no longer exist.

  1. b) The references included in the manuscript, with the exception of a few, are older than 5 years. Please enter the most recent references.

It is difficult to find recent references on this subject, but we have added one (9th reference).

  1. c) Authors, please add more references to the toxic effects of Oak Acorn; many symptoms have been listed, but all with a single reference.

References have been added.

  1. d) The authors are requested to expand the introduction.

We have developed the introduction.

Best regards,

Reviewer 3 Report

REVIEW:

Oak Acorn Poisoning In Cattle During Fall 2022: A Case Series and Review of the 2 Current Knowledge 3

Overall this paper is very well written, easy to follow and regarding an important topic.

General comments and feedback:

Title, Line 1 – consider replacing ‘fall’ with ‘autumn’ as it is a more universal term. Especially since autumn is the term used in the abstract.

Also, since in Lines 142-143 the authors mention there is no specific diagnostic test currently to confirm acorn poisoning, I suggest adding ‘Suspected Oak Acorn Poisoning’ to the title.

It might be worth highlighting in the summary and/or abstract that these cases arose from 3 separate farms because this further highlights this was not an isolated incidence, and hence more awareness is needed. The authors could also mention that suspected cases are reported every year.

Simple summary, Line 16 – exposure to which part of the oak? Only the acorns?

Line 23 – awareness should be raised – in any particular stakeholder groups? Just farmers? Attending veterinarians too?

Lines 23 – 24 – add that awareness is important given there is no known antidote to further stress that importance?

Line 25 – same question as for Line 16, more specificity relative to the nature of the oak poisoning.

Also, I know space is limited and this is for summary purposes only, but what does ‘slightly described’ mean – just an overview of potential risks? Case studies? A toxicological treatise?

Lines 35 – 37: ‘Animals are identified as sick lately, when renal failure is already established. Farmers should be made aware of the prevention of this disease, especially in years when acorns are abundant.’

I suggest rewording along the lines of: ‘Animals are identified as sick too late, when renal failure is already established. Farmers should be made more aware in order to prevent exposure, especially in years when acorns are abundant.’

Lines 43 – 44: Many parts of the oak, or many species of oak are toxic?

Line 56: not clear on what ‘has been set for a long time’? Does it mean that the repercussions have long been established/known?

Lines 74 - 75: add – ‘and the risk of exposure to acorns in a meadow must be emphasized and mitigated’

Lines 77 - 78: where exactly does the ‘our’ in ‘our grasslands’ refer to? And ‘full descriptions’ of what?

Lines 83-84: was the suspicion of oak poisoning on the part of the farmers? Who suspected oak poisoning?

Table 1 is great! What a fantastic resource for anyone trying to compare values, especially given the current dearth in information.

Lines 117-118: ‘or any other causes of renal impairment’…typically found or reported for livestock?

In Table 2, what exactly does LOW refer to? Falling outside detection levels?

Also, where there is a dash in the table, does it indicate non-detection or non-testing for that parameter? That seems to be the case based on reading Lines 123-130 but it would be helpful to indicate this in the table somehow as well to make it more standalone.

Perhaps instead of ‘economic reasons’ it might be clearer to say ‘financial constraints’

Figure 1, for increased clarity I suggest adding a slender colorized arrow (red, purple or green) pinpointing the exact location of the odoema, despite the clear caption

This type of arrow:

Table 3 – worth indicating there are no reported parameters in some places due to euthanasia of some individuals?

Figure 2, use more obvious arrows (see above) rather than the black triangles – same comment for Figure 3 although I like the colors!

Lines 221-222: Tens of litres of fluid? Or 10 litres of fluid? Would any fluid be acceptable?

How long between ingestion and initial onset of symptoms, and then once exposure has occurred, how long til demise

Lines 288-289: reword, ‘growth’ to ‘grew’

Lines 296-297: I suggest rewording this sentence: ‘Acorn poisoning depends on the year, the season, the age of the oak, the species, the maturity of the acorns and the weather conditions’

Along the lines of: ‘The prevalence of exposure to acorns and subsequent poisoning  depends on the year, the season, the age of the oak, the species, the maturity of the acorns and the weather conditions’

Lines 303-305: would it be difficult to quickly/approximately tally the number of poisonings reported, say in the last 5-8 years by LTGU? This could really strengthen the authors’ case.

Line 325: what is meant by ‘frustrating’ in this context?

Lines 358-360: the report on one cow where there is doubt about oak acorn poisoning diagnosis lends weight to using ‘suspected poisoning’.

Line 375: replace ‘this’ with ‘these’ chronic cases.

Line 383: to zebu add (Bos taurus indicus) – also, can the authors provide some very rough relative costs (hemiodialysis vs sale of a production animal) for reference?

General thoughts and questions:

Do the authors have any thoughts or ideas on parameters that could be used towards the development of a conclusive diagnostic test?

An appendix detailing more information about the individual cattle (age, sex) would be useful, especially for comparing those that survived and those that did not relative to, for example, the argument that younger more inquisitive animals are more susceptible to poisoning

I do understand the veterinary context of the term ‘diseas’. However, in this case is acorn poisoning in fact a disease, strictly speaking? Or is it not more accurate to say that – instead of acorn poisoning being an insidious disease – exposure to acorns can have severe repercussions? I think this is a way of framing the situation that remains accurate but that is clearer for more people to grasp that this can be prevented if known about.

The Conclusions section could culminate in the authors reiterating the need for greater awareness since this is inherently preventable and clearly, with cases still being reported annually, there remains a need for more awareness. Also the need, perhaps, for the development of a diagnostic test to conclusively implicate or rule out oak acorn poisoning, especially, for example, given the case of the one cow where there were co-occurring symptoms which introduced some doubt.

The English is absolutely fine, just a few adjustments here and there for clarity and grammatical correctness.

Author Response

REVIEWER 3

Overall this paper is very well written, easy to follow and regarding an important topic.

Dear Reviewer 3,

Thank you very much for your detailed review.

General comments and feedback:

Title, Line 1 – consider replacing ‘fall’ with ‘autumn’ as it is a more universal term. Especially since autumn is the term used in the abstract.

You're right, we've made the change.

Also, since in Lines 142-143 the authors mention there is no specific diagnostic test currently to confirm acorn poisoning, I suggest adding ‘Suspected Oak Acorn Poisoning’ to the title.

There is no diagnostic test on live animals, but we were able to diagnose 6/7 animals with certainty because we carried out an autopsy. The lesions observed on the kidneys are characteristic of acorn poisoning. We expect to retain the title.

It might be worth highlighting in the summary and/or abstract that these cases arose from 3 separate farms because this further highlights this was not an isolated incidence, and hence more awareness is needed. The authors could also mention that suspected cases are reported every year.

Yes, you're right. We have added this information to the summaries and the conclusion (line 425).

Simple summary, Line 16 – exposure to which part of the oak? Only the acorns?

No, we explain this in the introduction (line 46).

Line 23 – awareness should be raised – in any particular stakeholder groups? Just farmers? Attending veterinarians too?

For reasons of space, we have specified this in the abstract.

Lines 23 – 24 – add that awareness is important given there is no known antidote to further stress that importance?

We have added an on line 39.

Line 25 – same question as for Line 16, more specificity relative to the nature of the oak poisoning.

No, we explain this in the introduction (line 46).

Also, I know space is limited and this is for summary purposes only, but what does ‘slightly described’ mean – just an overview of potential risks? Case studies? A toxicological treatise?

There is little literature available on the subject.

Lines 35 – 37: ‘Animals are identified as sick lately, when renal failure is already established. Farmers should be made aware of the prevention of this disease, especially in years when acorns are abundant.’

I suggest rewording along the lines of: ‘Animals are identified as sick too late, when renal failure is already established. Farmers should be made more aware in order to prevent exposure, especially in years when acorns are abundant.’

The change has been made.

Lines 43 – 44: Many parts of the oak, or many species of oak are toxic?

Many species of. The change has been made on line 45.

Line 56: not clear on what ‘has been set for a long time’? Does it mean that the repercussions have long been established/known?

Above all, this means that there are not many reports of recent cases.

Lines 74 - 75: add – ‘and the risk of exposure to acorns in a meadow must be emphasized and mitigated’

This has been added in line 83.

Lines 77 - 78: where exactly does the ‘our’ in ‘our grasslands’ refer to? And ‘full descriptions’ of what?

Yes, this refers to our meadows.

We have added a full description of the case report (line 85).

Lines 83-84: was the suspicion of oak poisoning on the part of the farmers? Who suspected oak poisoning?

No, it's a suspicion on the part of the field vet. We specified this on line 93.

Table 1 is great! What a fantastic resource for anyone trying to compare values, especially given the current dearth in information.

Thanks for your comment!

Lines 117-118: ‘or any other causes of renal impairment’…typically found or reported for livestock?

Reported for livestock.

In Table 2, what exactly does LOW refer to? Falling outside detection levels?

Also, where there is a dash in the table, does it indicate non-detection or non-testing for that parameter? That seems to be the case based on reading Lines 123-130 but it would be helpful to indicate this in the table somehow as well to make it more standalone.

We've changed this to make it easier to understand.

Perhaps instead of ‘economic reasons’ it might be clearer to say ‘financial constraints’

The change has been made.

Figure 1, for increased clarity I suggest adding a slender colorized arrow (red, purple or green) pinpointing the exact location of the odoema, despite the clear caption

This type of arrow:

Unfortunately, the image of your arrow didn't appear on the website, so we've taken the liberty of adding an arrow. I hope you'll like it.

Table 3 – worth indicating there are no reported parameters in some places due to euthanasia of some individuals?

Yes, that's right.

Figure 2, use more obvious arrows (see above) rather than the black triangles – same comment for Figure 3 although I like the colors!

We've changed the arrows.

Lines 221-222: Tens of litres of fluid? Or 10 litres of fluid? Would any fluid be acceptable?

That's tens of liters. The exact number varied from animal to animal. It's not normal to find so much liquid in an abdominal cavity.

How long between ingestion and initial onset of symptoms, and then once exposure has occurred, how long til demise

As mentioned in point 2, it took between 3 weeks and a month before these animals, which had just fallen ill a few days earlier, were referred to us. After an initial unsuccessful treatment by their vet, they were all quickly referred. Then, in the case report, we can see that the animal that was euthanised the latest was 5 days out of hospital.

Lines 288-289: reword, ‘growth’ to ‘grew’

This has been changed.

Lines 296-297: I suggest rewording this sentence: ‘Acorn poisoning depends on the year, the season, the age of the oak, the species, the maturity of the acorns and the weather conditions’

Along the lines of: ‘The prevalence of exposure to acorns and subsequent poisoning  depends on the year, the season, the age of the oak, the species, the maturity of the acorns and the weather conditions’

This has been changed.

Lines 303-305: would it be difficult to quickly/approximately tally the number of poisonings reported, say in the last 5-8 years by LTGU? This could really strengthen the authors’ case.

We were unable to obtain the information.

Line 325: what is meant by ‘frustrating’ in this context?

Not very suggestive clinical signs. This has been changed in line 361.

Lines 358-360: the report on one cow where there is doubt about oak acorn poisoning diagnosis lends weight to using ‘suspected poisoning’.

We think our title is appropriate as long as it is clearly explained in the text. We hope you agree.

Line 375: replace ‘this’ with ‘these’ chronic cases.

This has been changed (line 411).

Line 383: to zebu add (Bos taurus indicus) – also, can the authors provide some very rough relative costs (hemiodialysis vs sale of a production animal) for reference?

We have added the species (line 419).

According to our information, a dialysis session costs between 750 and 1,000 euros, or a package of 6,000 euros for a medium-sized dog. It's wise to assume that it would be more expensive for a cattle, as the price is calculated according to the quantity of blood to be purified.

Fattened Belgian Blue cattle are sold to the slaughterhouse for around 2,000-3,000 euros. This represents the majority of cattle. Of course, individuals with exceptional genetics can sometimes be worth much more, and perhaps in such cases dialysis might be economically feasible. In our case, it wasn't interesting.

General thoughts and questions:

Do the authors have any thoughts or ideas on parameters that could be used towards the development of a conclusive diagnostic test?

Not really, and we haven't seen any literature on the subject. Maybe measuring tannin compounds in the blood, but there are several.

An appendix detailing more information about the individual cattle (age, sex) would be useful, especially for comparing those that survived and those that did not relative to, for example, the argument that younger more inquisitive animals are more susceptible to poisoning

We have this information in table 1, and find it more pleasant to read when it's included in the text. We hope you agree.

I do understand the veterinary context of the term ‘diseas’. However, in this case is acorn poisoning in fact a disease, strictly speaking? Or is it not more accurate to say that – instead of acorn poisoning being an insidious disease – exposure to acorns can have severe repercussions? I think this is a way of framing the situation that remains accurate but that is clearer for more people to grasp that this can be prevented if known about.

You're right, the term "disease" is difficult to use, so we've modified our text as advised.

The Conclusions section could culminate in the authors reiterating the need for greater awareness since this is inherently preventable and clearly, with cases still being reported annually, there remains a need for more awareness. Also the need, perhaps, for the development of a diagnostic test to conclusively implicate or rule out oak acorn poisoning, especially, for example, given the case of the one cow where there were co-occurring symptoms which introduced some doubt.

You're right, we've added these aspects to our conclusions.
